# MULTI-EPL:
# ACCURATE MULTI-SOURCE DOMAIN ADAPTATION

## ABSTRACT

*Given multiple source datasets with labels, how can we train a target model with no labeled data?* Multi-source domain adaptation (MSDA) aims to train a model using multiple source datasets different from a target dataset in the absence of target data labels. MSDA is a crucial problem applicable to many practical cases where labels for the target data are unavailable due to privacy issues. Existing MSDA frameworks are limited since they align data without considering conditional distributions $p(\mathbf{x}|y)$ of each domain. They also do not fully utilize the target data without labels, and rely on limited feature extraction with a single extractor. In this paper, we propose MULTI-EPL, a novel method for multi-source domain adaptation. MULTI-EPL exploits label-wise moment matching to align conditional distributions $p(\mathbf{x}|y)$, uses pseudolabels for the unavailable target labels, and introduces an ensemble of multiple feature extractors for accurate domain adaptation. Extensive experiments show that MULTI-EPL provides the state-of-the-art performance for multi-source domain adaptation tasks in both of image domains and text domains.

## 1 INTRODUCTION

*Given multiple source datasets with labels, how can we train a target model with no labeled data?* A large training data are essential for training deep neural networks. Collecting abundant data is unfortunately an obstacle in practice; even if enough data are obtained, manually labeling those data is prohibitively expensive. Using other available or much cheaper datasets would be a solution for these limitations; however, indiscriminate usage of other datasets often brings severe generalization error due to the presence of dataset shifts (Torralba & Efros (2011)). Unsupervised domain adaptation (UDA) tackles these problems where no labeled data from the target domain are available, but labeled data from other source domains are provided. Finding out domain-invariant features has been the focus of UDA since it allows knowledge transfer from the labeled source dataset to the unlabeled target dataset. There have been many efforts to transfer knowledge from a single source domain to a target one. Most recent frameworks minimize the distance between two domains by deep neural networks and distance-based techniques such as discrepancy regularizers (Long et al. (2015; 2016; 2017)), adversarial networks (Ganin et al. (2016); Tzeng et al. (2017)), and generative networks (Liu et al. (2017); Zhu et al. (2017); Hoffman et al. (2018b)).

While the above-mentioned approaches consider one single source, we address multi-source domain adaptation (MSDA), which is very crucial and more practical in real-world applications as well as more challenging. MSDA is able to bring significant performance enhancement by virtue of accessibility to multiple datasets as long as multiple domain shift problems are resolved. Previous works have extensively presented both theoretical analysis (Ben-David et al. (2010); Mansour et al. (2008); Crammer et al. (2008); Hoffman et al. (2018a); Zhao et al. (2018); Zellinger et al. (2020)) and models (Zhao et al. (2018); Xu et al. (2018); Peng et al. (2019)) for MSDA. MDAN (Zhao et al. (2018)) and DCTN (Xu et al. (2018)) build adversarial networks for each source domain to generate features domain-invariant enough to confound domain classifiers. However, these approaches do not encompass the shifts among source domains, counting only shifts between source and target domain. $M^3$SDA (Peng et al. (2019)) adopts moment matching strategy but makes the unrealistic assumption that matching the marginal probability $p(\mathbf{x})$ would guarantee the alignment of the conditional probability $p(\mathbf{x}|y)$. Most of these methods also do not fully exploit the knowledge of target

domain, imputing to the inaccessibility to the labels. Furthermore, all these methods leverage one single feature extractor, which possibly misses important information regarding label classification.

In this paper, we propose MULTI-EPL (Multi-source domain adaptation with Ensemble of feature extractors, Pseudolabels, and Label-wise moment matching), a novel MSDA framework which mitigates the limitations of these methods of not explicitly considering conditional probability $p(\mathbf{x}|y)$, and relying on only one feature extractor. The model architecture is illustrated in Figure 1. MULTI-EPL aligns the conditional probability $p(\mathbf{x}|y)$ by utilizing label-wise moment matching. We employ pseudolabels for the inaccessible target labels to maximize the usage of the target data. Moreover, generating an ensemble of features from multiple feature extractors gives abundant information about labels to the extracted features. Extensive experiments show the superiority of our methods.

Our contributions are summarized as follows:

- **Method.** We propose MULTI-EPL, a novel approach for MSDA that effectively obtains domain-invariant features from multiple domains by matching conditional probability $p(\mathbf{x}|y)$, utilizing pseudolabels for inaccessible target labels to fully deploy target data, and using an ensemble of multiple feature extractors. It allows domain-invariant features to be extracted, capturing the intrinsic differences of different labels.
- **Analysis.** We theoretically prove that minimizing the label-wise moment matching loss is relevant to bounding the target error.
- **Experiments.** We conduct extensive experiments on image and text datasets. We show that 1) MULTI-EPL provides the state-of-the-art accuracy, and 2) each of our main ideas significantly contributes to the superior performance.

## 2 RELATED WORK

**Single-source Domain Adaptation.** Given a labeled source dataset and an unlabeled target dataset, single-source domain adaptation aims to train a model that performs well on the target domain. The challenge of single-source domain adaptation is to reduce the discrepancy between the two domains and to obtain appropriate domain-invariant features. Various discrepancy measures such as Maximum Mean Discrepancy (MMD) (Tzeng et al. (2014); Long et al. (2015; 2016; 2017); Ghifary et al. (2016)) and KL divergence (Zhuang et al. (2015)) have been used as regularizers. Inspired from the insight that the domain-invariant features should exclude the clues about its domain, constructing adversarial networks against domain classifiers has shown superior performance. Liu et al. (2017) and Hoffman et al. (2018b) deploy GAN to transform data across the source and target domain, while Ganin et al. (2016) and Tzeng et al. (2017) leverage the adversarial networks to extract common features of the two domains. Unlike these works, we focus on *multiple* source domains.

**Multi-source Domain Adaptation.** Single-source domain adaptation should not be naively employed for multiple source domains due to the shifts between source domains. Many previous works have tackled MSDA problems theoretically. Mansour et al. (2008) establish distribution weighted combining rule that the weighted combination of source hypotheses is a good approximation for the target hypothesis. The rule is further extended to a stochastic case with joint distribution over the input and the output space in Hoffman et al. (2018a). Crammer et al. (2008) propose the general theory of how to sift appropriate samples out of multi-source data using expected loss. Efforts to find out transferable knowledge from multiple sources from the causal viewpoint are made in Zhang et al. (2015). There have been salient studies on the learning bounds for MSDA. Ben-David et al. (2010) found the generalization bounds based on $\mathcal{H}\Delta\mathcal{H}$-divergence, which are further tightened by Zhao et al. (2018). Frameworks for MSDA have been presented as well. Zhao et al. (2018) propose learning algorithms based on the generalization bounds for MSDA. DCTN (Xu et al. (2018)) resolves domain and category shifts between source and target domains via adversarial networks. $M^3$SDA (Peng et al. (2019)) associates all the domains into a common distribution by aligning the moments of the feature distributions of multiple domains. Lin et al. (2020) focus on the visual sentiment classification tasks and attempts to find out the common latent space of source and target domains. Wang et al. (2020) consider the interactions among multiple domains and reflect the information by constructing knowledge graph. However, all these methods do not consider multimode structures (Pei et al. (2018)) that differently labeled data follow distinct distributions, even if they are drawn from the same domain. Also, the domain-invariant features in these methods contain the label information for only one label classifier which lead these methods to miss a large amount of

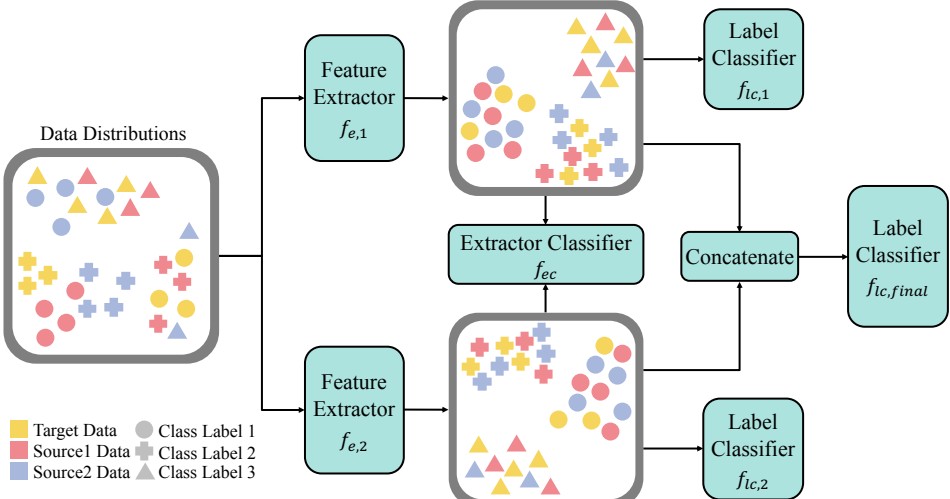

Figure 1: MULTI-EPL for $n$=2. MULTI-EPL consists of $n$ pairs of feature extractor and label classifier, one extractor classifier, and one final label classifier. Colors and symbols of the markers indicate domains and class labels of the data, respectively.

label information. Different from these methods, our frameworks fully count the multimodal structures handling the data distributions in a label-wise manner and minimize the label information loss considering multiple label classifiers.

**Moment Matching.** Domain adaptation has deployed the moment matching strategy to minimize the discrepancy between source and target domains. MMD regularizer (Tzeng et al. (2014); Long et al. (2015; 2016; 2017); Ghifary et al. (2016)) can be interpreted as the first-order moment while Sun et al. (2016) address second-order moments of source and target distributions. Zellinger et al. (2017) investigate the effect of higher-order moment matching. M$^3$SDA (Peng et al. (2019)) demonstrates that moment matching yields remarkable performance also with multiple sources. While previous works have focused on matching the moments of marginal distributions for single-source adaptation, we handle conditional distributions in multi-source scenarios.

## 3 PROPOSED METHOD

In this section, we describe our proposed method, MULTI-EPL. We first formulate the problem definition in Section 3.1. Then, we describe our main ideas in Section 3.2. Section 3.3 elaborates how to match label-wise moment with pseudolabels and Section 3.4 extends the approach by adding the concept of ensemble learning. Figure 1 shows the overview of MULTI-EPL.

### 3.1 PROBLEM DEFINITION

Given a set of labeled datasets from $N$ source domains $\mathcal{S}_1, \ldots, \mathcal{S}_N$ and an unlabeled dataset from a target domain $\mathcal{T}$, we aim to construct a model that minimizes test error on $\mathcal{T}$. We formulate source domain $\mathcal{S}_i$ as a tuple of the data distribution $\mu_{\mathcal{S}_i}$ on data space $\mathcal{X}$ and the labeling function $l_{\mathcal{S}_i} \colon \mathcal{S}_i = (\mu_{\mathcal{S}_i}, l_{\mathcal{S}_i})$. Source dataset drawn with the distribution $\mu_{\mathcal{S}_i}$ is denoted as $\mathbf{X}_{\mathcal{S}_i} = \{(\mathbf{x}_j^{\mathcal{S}_i}, y_j^{\mathcal{S}_i})\}_{j=1}^{n_{\mathcal{S}_i}}$. Likewise, the target domain and the target dataset are denoted as $\mathcal{T} = (\mu_{\mathcal{T}}, l_{\mathcal{T}})$ and $\mathbf{X}_{\mathcal{T}} = \{\mathbf{x}_j^{\mathcal{T}}\}_{j=1}^{n_{\mathcal{T}}}$, respectively. We narrow our focus down to homogeneous settings in classification tasks: all domains share the same data space $\mathcal{X}$ and label set $\mathcal{C}$.

### 3.2 OVERVIEW

We propose MULTI-EPL based on the following observations: 1) existing methods focus on aligning the marginal distributions $p(\mathbf{x})$ not the conditional ones $p(\mathbf{x}|y)$, 2) knowledge of the target data is not fully employed as no target label is given, and 3) there exists a large amount of label information loss since domain-invariant features are extracted for only one label classifier. Thus, we design MULTI-EPL aiming to solve the limitations. Designing such method entails the following challenges:

1. **Matching conditional distributions.** How can we align the conditional distribution, $p(\mathbf{x}|y)$, of multiple domains not the marginal one, $p(\mathbf{x})$?
2. **Exploitation of the target data.** How can we fully exploit the knowledge of the target data despite the absence of the target labels?
3. **Maximally utilizing feature information.** How can we maximally utilize the information that the domain-invariant features contain?

We propose the following main ideas to address the challenges:

1. **Label-wise moment matching (Section 3.3).** We match the *label-wise* moments of the domain-invariant features so that the features with the same labels have similar distributions regardless of their original domains.
2. **Pseudolabels (Section 3.3).** We use pseudolabels as alternatives to the target labels.
3. **Ensemble of feature representations (Section 3.4).** We learn to extract ensemble of features from multiple feature extractors, each of which involves distinct domain-invariant features for its own label classifier.

## 3.3 LABEL-WISE MOMENT MATCHING WITH PSEUDOLABELS

We describe how MULTI-EPL matches conditional distributions $p(\mathbf{x}|y)$ of the features from multiple distinct domains. In MULTI-EPL, a feature extractor $f_e$ and a label classifier $f_{lc}$ lead the features to be domain-invariant and label-informative at the same time. The feature extractor $f_e$ extracts features from data, and the label classifier $f_{lc}$ receives the features and predicts the labels for the data. We train $f_e$ and $f_{lc}$, according to the losses for *label-wise moment matching* and *label classification*, which make the features domain-invariant and label-informative, respectively.

**Label-wise Moment Matching.** To achieve the alignment of domain-invariant features, we define a label-wise moment matching loss as follows:

$$\mathcal{L}_{lmm,K} = \frac{1}{|\mathcal{C}|} \binom{N+1}{2}^{-1} \sum_{k=1}^{K} \sum_{\mathcal{D},\mathcal{D}'} \sum_{c \in \mathcal{C}} \left\| \frac{1}{n_{\mathcal{D},c}} \sum_{j; y_j^{\mathcal{D}}=c} f_e(\mathbf{x}_j^{\mathcal{D}})^k - \frac{1}{n_{\mathcal{D}',c}} \sum_{j; y_j^{\mathcal{D}'}=c} f_e(\mathbf{x}_j^{\mathcal{D}'})^k \right\|_2, \quad (1)$$

where $K$ is a hyperparameter indicating the maximum order of moments considered by the loss, $\mathcal{D}$ and $\mathcal{D}'$ are two distinct domains amongst the $N+1$ domains, and $n_{\mathcal{D},c}$ is the number of data labeled as $c$ in $\mathbf{X}_{\mathcal{D}}$. We introduce *pseudolabels* for the target data, which are determined by the outputs of the model currently being trained, to manage the absence of the ground truths for the target data. In other words, we leverage $f_{lc}(f_e(\mathbf{x}^{\mathcal{T}}))$ to give the pseudolabel to the target data $\mathbf{x}^{\mathcal{T}}$. Drawing the pseudolabels using the incomplete model, however, brings mis-labeling issue which impedes further training. To alleviate this problem, we set a threshold $\tau$ and assign the pseudolabels to the target data only when the prediction confidence is greater than the threshold. The target examples with low confidence are not pseudolabeled and not counted in label-wise moment matching.

By minimizing $\mathcal{L}_{lmm,K}$, the feature extractor $f_e$ aligns data from multiple domains by bringing consistency in distributions of the features with the same labels. The data with distinct labels are aligned independently, taking account of the multimode structures that differently labeled data follow different distributions.

**Label Classification.** The label classifier $f_{lc}$ gets the features projected by $f_e$ as inputs and makes the label predictions. The *label classification loss* is defined as follows:

$$\mathcal{L}_{lc} = \frac{1}{N} \sum_{i=1}^{N} \frac{1}{n_{\mathcal{S}_i}} \sum_{j=1}^{n_{\mathcal{S}_i}} \mathcal{L}_{ce}(f_{lc}(f_e(\mathbf{x}_j^{\mathcal{S}_i})), y_j^{\mathcal{S}_i}), \quad (2)$$

where $\mathcal{L}_{ce}$ is the softmax cross-entropy loss. Minimizing $\mathcal{L}_{lc}$ separates the features with different labels so that each of them gets label-distinguishable.

## 3.4 ENSEMBLE OF FEATURE REPRESENTATIONS

In this section, we introduce ensemble learning for further enhancement. Features extracted with the strategies elaborated in the previous section contain the label information for a single label classifier. However, each label classifier leverages only limited label characteristics, and thus the conventional scheme to adopt only one pair of feature extractor and label classifier captures only a small part of

the label information. Our idea is to leverage an ensemble of multiple pairs of feature extractor and label classifier in order to make the features to be more label-informative.

We train multiple pairs of feature extractor and label classifier in parallel following the label-wise moment matching approach explained in Section 3.3. Let $n$ denote the number of the feature extractors in the overall model. We denote the $n$ (feature extractor, label classifier) pairs as $(f_{e,1}, f_{lc,1}), (f_{e,2}, f_{lc,2}), \ldots, (f_{e,n}, f_{lc,n})$ and the $n$ resultant features as $feat_1, feat_2, \ldots, feat_n$ where $feat_i$ is the output of the feature extractor $f_{e,i}$. After obtaining $n$ different feature mapping modules, we concatenate the $n$ features into one vector $feat_{final} = concat(feat_1, feat_2, \ldots, feat_n)$. The final label classifier $f_{lc,final}$ takes the concatenated feature as input, and predicts the label of the feature.

Naively exploiting multiple feature extractors, however, does not guarantee the diversity of the features since it resorts to the randomness. Thus, we introduce a new model component, *extractor classifier*, which separates the features from different extractors. The extractor classifier $f_{ec}$ gets the features generated by a feature extractor as inputs and predicts which feature extractor has generated the features. For example, if $n = 2$, the extractor classifier $f_{ec}$ attempts to predict whether the input feature is extracted by the extractor $f_{e,1}$ or $f_{e,2}$. By training the extractor classifier and multiple feature extractors at once, we explicitly diversify the features obtained from different extractors. We train the extractor classifier utilizing the *feature diversifying loss*, $\mathcal{L}_{fd}$:

$$\mathcal{L}_{fd} = \frac{1}{N+1} \left( \sum_{i=1}^{N} \frac{1}{n_{\mathcal{S}_i}} \sum_{j=1}^{n_{\mathcal{S}_i}} \sum_{k=1}^{n} \mathcal{L}_{ce}(f_{e,k}(\mathbf{x}_j^{\mathcal{S}_i}), k) + \frac{1}{n_{\mathcal{T}}} \sum_{j=1}^{n_{\mathcal{T}}} \sum_{k=1}^{n} \mathcal{L}_{ce}(f_{e,k}(\mathbf{x}_j^{\mathcal{T}}), k) \right), \quad (3)$$

where $n$ is the number of feature extractors.

### 3.5 MULTI-EPL: ACCURATE MULTI-SOURCE DOMAIN ADAPTATION

Our final model MULTI-EPL consists of $n$ pairs of feature extractor and label classifier, $(f_{e,1}, f_{lc,1}), (f_{e,2}, f_{lc,2}), \ldots, (f_{e,n}, f_{lc,n})$, one extractor classifier $f_{ec}$, and one final label classifier $f_{lc,final}$. We first train the entire model except the final label classifier with the loss $\mathcal{L}$:

$$\mathcal{L} = \sum_{k=1}^{n} \mathcal{L}_{lc,k} + \alpha \sum_{k=1}^{n} \mathcal{L}_{lmm,K,k} + \beta \mathcal{L}_{fd}, \quad (4)$$

where $\mathcal{L}_{lc,k}$ is the label classification loss of the classifier $f_{lc,k}$, $\mathcal{L}_{lmm,K,k}$ is the label-wise moment matching loss of the feature extractor $f_{e,k}$, and $\alpha$ and $\beta$ are the hyperparameters. Then, the final label classifier is trained with respect to the label classification loss $\mathcal{L}_{lc,final}$ using the concatenated features from multiple feature extractors.

## 4 ANALYSIS

We present a theoretical insight regarding the validity of the label-wise moment matching loss. For simplicity, we tackle only binary classification tasks. The error rate of a hypothesis $h$ on a domain $\mathcal{D}$ is denoted as $\epsilon_{\mathcal{D}}(h) = Pr\{h(\mathbf{x}) \neq l_{\mathcal{D}}(\mathbf{x})\}$ where $l_{\mathcal{D}}$ is the labeling function on the domain $\mathcal{D}$. We first introduce $k$-th order label-wise moment divergence.

**Definition 1.** *Let $\mathcal{D}$ and $\mathcal{D}'$ be two domains over an input space $\mathcal{X} \subset \mathbb{R}^n$ where $n$ is the dimension of the inputs. Let $\mathcal{C}$ be the set of the labels, and $\mu_c(\mathbf{x})$ and $\mu'_c(\mathbf{x})$ be the data distribution given that the label is $c$, i.e. $\mu_c(\mathbf{x}) = \mu(\mathbf{x}|y = c)$ and $\mu'_c(\mathbf{x}) = \mu'(\mathbf{x}|y = c)$ for the data distribution $\mu$ and $\mu'$ on the domains $\mathcal{D}$ and $\mathcal{D}'$, respectively. Then, the $k$-th order label-wise moment divergence $d_{LM,k}(\mathcal{D}, \mathcal{D}')$ of the two domains $\mathcal{D}$ and $\mathcal{D}'$ over $\mathcal{X}$ is defined as*

$$d_{LM,k}(\mathcal{D}, \mathcal{D}') = \sum_{c \in \mathcal{C}} \sum_{\mathbf{i} \in \Delta_k} \left| p(c) \int_{\mathcal{X}} \mu_c(\mathbf{x}) \prod_{j=1}^{n} (x_j)^{i_j} d\mathbf{x} - p'(c) \int_{\mathcal{X}} \mu'_c(\mathbf{x}) \prod_{j=1}^{n} (x_j)^{i_j} d\mathbf{x} \right|, \quad (5)$$

*where $\Delta_k = \{\mathbf{i} = (i_1, \ldots, i_n) \in \mathbb{N}_0^n | \sum_{j=1}^{n} i_j = k\}$ is the set of the tuples of the nonnegative integers, which add up to $k$, $p(c)$ and $p'(c)$ are the probability that arbitrary data from $\mathcal{D}$ and $\mathcal{D}'$ to be labeled as $c$ respectively, and the data $\mathbf{x} \in \mathcal{X}$ is expressed as $(x_1, \ldots, x_n)$.* $\square$

The ultimate goal of MSDA is to find a hypothesis $h$ with the minimum target error. We nevertheless train the model with respect to the source data since ground truths for the target are unavailable. Let

$N$ datasets be drawn from $N$ labeled source domains $\mathcal{S}_1, \ldots, \mathcal{S}_N$ respectively. We denote $i$-th source dataset $\mathbf{X}_{\mathcal{S}_i}$ as $\{(\mathbf{x}_j^{\mathcal{S}_i}, y_j^{\mathcal{S}_i})\}_{j=1}^{n_{\mathcal{S}_i}}$. The empirical error of hypothesis $h$ in $i$-th source domain $\mathcal{S}_i$ estimated with $\mathbf{X}_{\mathcal{S}_i}$ is formulated as $\hat{\epsilon}_{\mathcal{S}_i}(h) = \frac{1}{n_{\mathcal{S}_i}} \sum_{j=1}^{n_{\mathcal{S}_i}} \mathbf{1}_{h(\mathbf{x}_j^{\mathcal{S}_i}) \neq y_j^{\mathcal{S}_i}}$. Given a weight vector $\boldsymbol{\alpha} = (\alpha_1, \alpha_2, \ldots, \alpha_N)$ such that $\sum_{i=1}^{N} \alpha_i = 1$, the weighted empirical source error is formulated as $\hat{\epsilon}_{\boldsymbol{\alpha}}(h) = \sum_{i=1}^{N} \alpha_i \hat{\epsilon}_{\mathcal{S}_i}(h)$. We extend the theorems in Ben-David et al. (2010); Peng et al. (2019) and derive a bound for the target error $\epsilon_{\mathcal{T}}(h)$, for $h$ trained with source data, in terms of $k$-th order label-wise moment divergence.

**Theorem 1.** *Let $\mathcal{H}$ be a hypothesis space of VC dimension $d$, $n_{\mathcal{S}_i}$ be the number of samples from source domain $\mathcal{S}_i$, $m = \sum_{i=1}^{N} n_{\mathcal{S}_i}$ be the total number of samples from $N$ source domains $\mathcal{S}_1, \ldots, \mathcal{S}_N$, and $\boldsymbol{\beta} = (\beta_1, \ldots, \beta_N)$ with $\beta_i = \frac{n_{\mathcal{S}_i}}{m}$. Let us define a hypothesis $\hat{h} = \arg\min_{h \in \mathcal{H}} \hat{\epsilon}_{\boldsymbol{\alpha}}(h)$ that minimizes the weighted empirical source error, and a hypothesis $h_{\mathcal{T}}^* = \arg\min_{h \in \mathcal{H}} \epsilon_{\mathcal{T}}(h)$ that minimizes the true target error. Then, for any $\delta \in (0,1)$ and $\epsilon > 0$, there exist $N$ integers $n_\epsilon^1, \ldots, n_\epsilon^N$ and $N$ constants $a_{n_\epsilon^1}, \ldots, a_{n_\epsilon^N}$ such that*

$$\epsilon_{\mathcal{T}}(\hat{h}) \leq \epsilon_{\mathcal{T}}(h_{\mathcal{T}}^*) + \eta_{\boldsymbol{\alpha},\boldsymbol{\beta},m,\delta} + \epsilon + \sum_{i=1}^{N} \alpha_i \left( 2\lambda_i + a_{n_\epsilon^i} \sum_{k=1}^{n_\epsilon^i} d_{LM,k}(\mathcal{S}_i, \mathcal{T}) \right) \tag{6}$$

*with probability at least $1 - \delta$, where $\eta_{\boldsymbol{\alpha},\boldsymbol{\beta},m,\delta} = 4\sqrt{\left( \sum_{i=1}^{N} \frac{\alpha_i^2}{\beta_i} \right) \left( \frac{2d\left( \log\left( \frac{2m}{d} \right) + 1 \right) + 2\log\left( \frac{4}{\delta} \right)}{m} \right)}$ and $\lambda_i = \min_{h \in \mathcal{H}} \{ \epsilon_{\mathcal{T}}(h) + \epsilon_{\mathcal{S}_i}(h) \}$.* □

*Proof.* See the Appendix A.1. □

Speculating that all datasets are balanced against the annotations, *i.e.*, $p(c) = p'(c) = \frac{1}{|\mathcal{C}|}$ for any $c \in \mathcal{C}$, $\mathcal{L}_{lmm,K}$ is expressed as the sum of the estimates of $d_{LM,k}$ with $k = 1, \ldots, K$. The theorem provides an insight that label-wise moment matching allows the model trained with source data to have performance comparable to the optimal one on the target domain.

## 5 EXPERIMENTS

We conduct experiments to answer the following questions of MULTI-EPL.

**Q1 Accuracy (Section 5.2).** How well does MULTI-EPL perform in classification tasks?

**Q2 Ablation Study (Section 5.3).** How much does each component of MULTI-EPL contribute to performance improvement?

**Q3 Effects of Degree of Ensemble (Section 5.4).** How does the performance change as the number $n$ of the pairs of the feature extractor and the label classifier increases?

### 5.1 EXPERIMENTAL SETTINGS

**Datasets.** We use three kinds of datasets, Digits-Five, Office-Caltech10[1], and Amazon Reviews[2]. Digits-Five consists of five datasets for digit recognition: MNIST[3] (LeCun et al. (1998)), MNIST-M[4] (Ganin & Lempitsky (2015)), SVHN[5] (Netzer et al. (2011)), SynthDigits[6] (Ganin & Lempitsky (2015)), and USPS[7] (Hastie et al. (2001)). We set one of them as a target domain and the rest as source domains. Following the conventions in prior works (Xu et al. (2018); Peng et al. (2019)), we randomly sample 25000 instances from the source training set and 9000 instances from the target training set to train the model except for USPS for which the whole training set is used.

---

[1] https://people.eecs.berkeley.edu/~jhoffman/domainadapt/
[2] https://github.com/KeiraZhao/MDAN/blob/master/amazon.npz
[3] http://yann.lecun.com/exdb/mnist/
[4] http://yaroslav.ganin.net
[5] http://ufldl.stanford.edu/housenumbers/
[6] http://yaroslav.ganin.net
[7] https://www.kaggle.com/bistaumanga/usps-dataset

Table 1: Summary of datasets.

|  | Datasets | Features | Labels | Training set | Test set | Properties |
|---|---|---|---|---|---|---|
| **Digits-Five** | MNIST | 1x28x28 | 10 | 60000 | 10000 | Grayscale images |
|  | MNIST-M | 3x32x32 | 10 | 59001 | 9001 | RGB images |
|  | SVHN | 3x32x32 | 10 | 73257 | 26032 | RGB images |
|  | SynthDigits | 3x32x32 | 10 | 479400 | 9553 | RGB images |
|  | USPS | 1x16x16 | 10 | 7291 | 2007 | Grayscale images |
| **Office-Caltech10** | Amazon | 3x300x300 | 10 | 958 | 958 | RGB images |
|  | Caltech | Variable | 10 | 1123 | 1123 | RGB images |
|  | DSLR | 3x1000x1000 | 10 | 157 | 157 | RGB images |
|  | Webcam | Variable | 10 | 295 | 295 | RGB images |
| **Amazon Reviews** | Books | 5000 | 2 | 2000 | 4465 | 5000-dim vector |
|  | DVDs | 5000 | 2 | 2000 | 3586 | 5000-dim vector |
|  | Electronics | 5000 | 2 | 2000 | 5681 | 5000-dim vector |
|  | Kitchen appliances | 5000 | 2 | 2000 | 5945 | 5000-dim vector |

Office-Caltech10 is the dataset for image classification with 10 categories that Office31 dataset and Caltech dataset have in common. It involves four different domains: Amazon, Caltech, DSLR, and Webcam. We double the number of data by data augmentation and exploit all the original data and augmented data as training data and test data respectively.Amazon Reviews dataset contains customers' reviews on 4 product categories: Books, DVDs, Electronics, and Kitchen appliances. The instances are encoded into 5000-dimensional vectors and are labeled as being either positive or negative depending on their sentiments. We set each of the four categories as a target and the rest as sources. For all the domains, 2000 instances are sampled for training, and the rest of the data are used for the test. Details about the datasets are summarized in Table 1.

**Competitors.** We use 3 MSDA algorithms, DCTN (Xu et al. (2018)), M³SDA (Peng et al. (2019)), and M³SDA-$\beta$ (Peng et al. (2019)), with state-of-the-art performances as baselines. All the frameworks share the same architecture for the feature extractor, the domain classifier, and the label classifier for consistency. For Digits-Five, we use convolutional neural networks based on LeNet5 (LeCun et al. (1998)). For Office-Caltech10, ResNet50 (He et al. (2016)) pretrained on ImageNet is used as the backbone architecture. For Amazon Reviews, the feature extractor is composed of three fully-connected layers each with 1000, 500, and 100 output units, and a single fully-connected layer with 100 input units and 2 output units is adopted for both of the extractor and label classifiers. With Digits-Five, LeNet5 (LeCun et al. (1998)) and ResNet14 (He et al. (2016)) without any adaptation are additionally investigated in two different manners: *Source Combined* and *Single Best*. In *Source Combined*, multiple source datasets are simply combined and fed into a model. In *Single Best*, we train the model with each source dataset independently, and report the result of the best performing one. Likewise, ResNet50 and MLP consisting of 4 fully-connected layers with 1000, 500, 100, and 2 units are investigated without adaptation for Office-Caltech10 and Amazon Reviews, respectively.

**Training Details.** We train our models for Digits-Five with Adam optimizer (Kingma & Ba (2015)) with $\beta_1 = 0.9$, $\beta_2 = 0.999$, and the learning rate of 0.0004 for 100 epochs. All images are scaled to $32 \times 32$ and the mini batch size is set to 128. We set the hyperparameters $\alpha = 0.0005$, $\beta = 1$, and $K = 2$. For the experiments with Office-Caltech10, all the modules comprising our model are trained following SGD with the learning rate 0.001, except that the optimizers for feature extractors have the learning rate 0.0001. We scale all the images to $224 \times 224$ and set the mini batch size to 48. All the hyperparameters are kept the same as in the experiments with Digits-Five. For Amazon Reviews, we train the models for 50 epochs using Adam optimizer with $\beta_1 = 0.9$, $\beta_2 = 0.999$, and the learning rate of 0.0001. We set $\alpha = \beta = 1$, $K = 2$, and the mini batch size to 100. For every experiment, the confidence threshold $\tau$ is set to 0.9.

## 5.2 PERFORMANCE EVALUATION

We evaluate the performance of MULTI-EPL with $n = 2$ against the competitors. We repeat experiments for each setting five times and report the mean and the standard deviation. The results are summarized in Table 2. Note that MULTI-EPL provides the best accuracy in all the datasets, showing its consistent superiority in both image datasets (Digits-Five, Office-Caltech10) and text dataset (Amazon Reviews). The enhancement is remarkable especially when MNIST-M is the target domain in Digits-Five, improving the accuracy by 11.48% compared to the state-of-the-art methods.

Table 2: Classification accuracy on Digits-Five, Office-Caltech10, and Amazon Reviews with and without domain adaptation. The letters before and after the slash represent source domains and a target domain respectively. In Digits-Five, T, M, S, D, and U stands for MNIST, MNIST-M, SVHN, SynthDigits, and USPS respectively. In Office-Caltech10 and Amazon Reviews, we indicate each domain using the first letter of its name. SC and SB indicate *Source Combined* and *Single Best*, respectively. Note that MULTI-EPL shows the best performance.

(a) Digits-Five

| Method | M+S+D+U/T | T+S+D+U/M | T+M+D+U/S | T+M+S+U/D | T+M+S+D/U | Average |
|---|---|---|---|---|---|---|
| LeNet5 (SC) | 97.58±0.18 | 61.72±1.38 | 75.15±0.76 | 80.29±0.66 | 81.58±1.51 | 79.27±0.90 |
| ResNet14 (SC) | 98.22±0.26 | 63.53±0.84 | 79.08±1.63 | 92.85±0.48 | 94.51±0.31 | 85.64±0.70 |
| LeNet5 (SB) | 97.09±0.14 | 51.10±1.87 | 76.75±0.57 | 79.92±0.50 | 83.28±0.92 | 77.63±0.80 |
| ResNet14 (SB) | 97.07±1.03 | 49.48±1.30 | 81.40±0.70 | 91.79±0.53 | 91.54±2.68 | 82.33±1.25 |
| DCTN | 99.28±0.06 | 71.99±1.58 | 78.34±1.10 | 91.55±0.65 | 98.43±0.23 | 87.92±0.72 |
| M³SDA | 98.75±0.05 | 67.77±0.71 | 81.75±0.61 | 88.51±0.29 | 97.17±0.22 | 86.79±0.38 |
| M³SDA-$\beta$ | 98.99±0.03 | 72.47±0.19 | 81.40±0.28 | 89.51±0.37 | 97.40±0.19 | 87.95±0.21 |
| MULTI-EPL (n=2) | **99.31±0.04** | **83.95±0.90** | **86.93±0.39** | **93.15±0.17** | **98.49±0.08** | **92.37±0.31** |

(b) Office-Caltech10

| Method | C+D+W/A | A+D+W/C | A+C+W/D | A+C+D/W | Average |
|---|---|---|---|---|---|
| ResNet50 (SC) | 95.47±0.25 | 91.59±0.51 | 99.36±0.78 | 99.26±0.37 | 96.42±0.48 |
| ResNet50 (SB) | 95.03±0.48 | 89.05±0.88 | 99.87±0.28 | 98.24±0.61 | 95.55±0.56 |
| DCTN | 95.05±0.24 | 90.60±0.71 | **100.0±0.00** | 99.46±0.62 | 96.28±0.39 |
| M³SDA | 95.14±0.31 | 93.59±0.40 | 99.49±0.53 | **99.86±0.19** | 97.02±0.36 |
| M³SDA-$\beta$ | 94.36±0.26 | 91.70±0.71 | 99.75±0.35 | 99.39±0.15 | 96.30±0.37 |
| MULTI-EPL (n=2) | **95.74±0.29** | **93.91±0.28** | 99.87±0.28 | **99.86±0.19** | **97.35±0.26** |

(c) Amazon Reviews

| Method | D+E+K/B | B+E+K/D | B+D+K/E | B+D+E/K | Average |
|---|---|---|---|---|---|
| MLP (SC) | 79.76±0.70 | 82.18±0.59 | 84.42±0.27 | 87.23±0.51 | 83.40±0.52 |
| MLP (SB) | 79.00±0.92 | 80.38±0.61 | 84.76±0.45 | 87.46±0.36 | 82.90±0.58 |
| DCTN | 78.92±0.56 | 81.22±1.01 | 83.56±1.52 | 86.47±0.71 | 82.54±0.95 |
| M³SDA | 78.97±0.79 | 80.51±0.99 | 83.63±0.68 | 85.99±0.85 | 82.27±0.83 |
| M³SDA-$\beta$ | 80.26±0.43 | 81.80±0.72 | 85.02±0.34 | 86.99±0.56 | 83.52±0.51 |
| MULTI-EPL (n=2) | **81.14±0.29** | **83.13±0.45** | **86.47±0.35** | **88.53±0.33** | **84.82±0.35** |

## 5.3 ABLATION STUDY

We perform an ablation study on Digits-Five to identify what exactly enhances the performance of MULTI-EPL. We compare MULTI-EPL with 3 of its variants: MULTI-0, MULTI-PL, and MULTI-EPL-R. MULTI-0 aligns moments regardless of the labels of the data. MULTI-PL trains the model without ensemble learning. MULTI-EPL-R exploits ensemble learning strategy but relies on randomness without the extractor classifier and the feature diversifying loss.

The results are shown in Table 3. By comparing MULTI-0 with MULTI-PL, we observe that considering labels in moment matching plays a significant role in extracting domain-invariant features. The remarkable performance gap between MULTI-PL and MULTI-EPL with $n = 2$ verifies the effectiveness of ensemble learning. Comparing MULTI-EPL and MULTI-EPL-R, MULTI-EPL shows a better performance than MULTI-EPL-R in half of the cases; this means that explicitly diversifying loss often helps further improve the accuracy, while resorting to randomness for feature diversification also works in general. Hence, we conclude that we are able to apply ensemble learning approach without concern about the redundancy in features.

## 5.4 EFFECTS OF ENSEMBLE

We vary $n$, the number of pairs of feature extractor and label classifier, and repeat the performance evaluation on Digits-Five. The results are summarized in Table 3. While an ensemble of two pairs gives much better performance than the model with a single pair, using more than two pairs rarely brings further improvement. This result demonstrates that two pairs of feature extractor and label classifier are able to cover most information without losing important label information in Digits-Five. It is notable that increasing $n$ sometimes brings small performance degradation. As more feature extractors are adopted to obtain final features, the complexity of final features increases. It

Table 3: Experiments with MULTI-EPL and its variants.

| Method | M+S+D+U/T | T+S+D+U/M | T+M+D+U/S | T+M+S+U/D | T+M+S+D/U | Average |
|---|---|---|---|---|---|---|
| MULTI-0 | 98.75±0.05 | 67.77±0.71 | 81.75±0.61 | 88.51±0.29 | 97.17±0.22 | 86.79±0.38 |
| MULTI-PL | 99.14±0.06 | 79.32±0.73 | 84.77±0.39 | 91.91±0.05 | 98.49±0.16 | 90.73±0.28 |
| MULTI-EPL-R (n=2) | **99.34±0.05** | 83.24±0.81 | 86.96±0.34 | 92.88±0.15 | **98.56±0.17** | 92.20±0.30 |
| MULTI-EPL (n=2) | 99.31±0.04 | **83.95±0.90** | 86.93±0.39 | **93.15±0.17** | 98.49±0.08 | **92.37±0.31** |
| MULTI-EPL (n=3) | 99.31±0.05 | 82.78±0.67 | **87.10±0.29** | 92.85±0.24 | 98.48±0.09 | 92.10±0.27 |
| MULTI-EPL (n=4) | 99.30±0.07 | 82.74±0.55 | 86.65±0.41 | 92.86±0.15 | 98.50±0.08 | 92.01±0.25 |

is harder for the final label classifier to manage the features with high complexity compared to the simple ones. This deteriorates the performance when we exploit more than two feature extractors.

## 6 CONCLUSION

We propose MULTI-EPL, a novel framework for the multi-source domain adaptation problem. MULTI-EPL overcomes the problems in existing methods of not directly addressing conditional distributions of data $p(\mathbf{x}|y)$, not fully exploiting knowledge of target data, and missing large amount of label information. MULTI-EPL aligns data from multiple source domains and the target domain considering the data labels, and exploits pseudolabels for exploiting unlabeled target data. MULTI-EPL further enhances the performance by generating an ensemble of multiple feature extractors. Our framework exhibits superior performance on both image and text classification tasks. Considering labels in moment matching and adding ensemble learning idea is shown to bring remarkable performance enhancement through ablation study. Future works include extending our approach to other tasks such as regression, which may require modification in the pseudolabeling method.

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

## A  APPENDIX

### A.1  PROOF FOR THEOREM 1

In this section, we prove Theorem 1 in the paper. We first define $k$-th order label-wise moment divergence $d_{LM,k}$, and disagreement ratio $\epsilon_{\mathcal{D}}(h_1, h_2)$ of the two hypotheses $h_1, h_2 \in \mathcal{H}$ on the domain $\mathcal{D}$.

**Definition 1.** *Let $\mathcal{D}$ and $\mathcal{D}'$ be two domains over an input space $\mathcal{X} \subset \mathbb{R}^n$ where $n$ is the dimension of the inputs. Let $\mathcal{C}$ be the set of the labels, and $\mu_c(\mathbf{x})$ and $\mu'_c(\mathbf{x})$ be the data distributions given that the label is $c$, i.e. $\mu_c(\mathbf{x}) = \mu(\mathbf{x}|y = c)$ and $\mu'_c(\mathbf{x}) = \mu'(\mathbf{x}|y = c)$ for the data distribution $\mu$ and $\mu'$ on the domains $\mathcal{D}$ and $\mathcal{D}'$, respectively. Then, the $k$-th order label-wise moment divergence $d_{LM,k}(\mathcal{D}, \mathcal{D}')$ of the two domains $\mathcal{D}$ and $\mathcal{D}'$ over $\mathcal{X}$ is defined as*

$$d_{LM,k}(\mathcal{D}, \mathcal{D}') = \sum_{c \in \mathcal{C}} \sum_{\mathbf{i} \in \Delta_k} \left| p(c) \int_{\mathcal{X}} \mu_c(\mathbf{x}) \prod_{j=1}^{n} (x_j)^{i_j} d\mathbf{x} - p'(c) \int_{\mathcal{X}} \mu'_c(\mathbf{x}) \prod_{j=1}^{n} (x_j)^{i_j} d\mathbf{x} \right|, \quad (7)$$

*where $\Delta_k = \{\mathbf{i} = (i_1, \ldots, i_n) \in \mathbb{N}_0^n | \sum_{j=1}^{n} i_j = k\}$ is the set of the tuples of the nonnegative integers, which add up to $k$, $p(c)$ and $p'(c)$ are the probability that arbitrary data from $\mathcal{D}$ and $\mathcal{D}'$ to be labeled as $c$ respectively, and the data $\mathbf{x} \in \mathcal{X}$ is expressed as $(x_1, \ldots, x_n)$.* □

**Definition 2.** *Let $\mathcal{D}$ be a domain over an input space $\mathcal{X} \subset \mathbb{R}^n$ with the data distribution $\mu(\mathbf{x})$. Then, we define the disagreement ratio $\epsilon_{\mathcal{D}}(h_1, h_2)$ of the two hypotheses $h_1, h_2 \in \mathcal{H}$ on the domain $\mathcal{D}$ as*

$$\epsilon_{\mathcal{D}}(h_1, h_2) = \Pr_{\mathbf{x} \sim \mu(\mathbf{x})} [h_1(\mathbf{x}) \neq h_2(\mathbf{x})]. \quad (8)$$

□

**Theorem 2.** *(Stone-Weierstrass Theorem (Stone (1937))) Let $K$ be a compact subset of $\mathbb{R}^n$ and $f : K \to \mathbb{R}$ be a continuous function. Then, for every $\epsilon > 0$, there exists a polynomial, $P : K \to \mathbb{R}$, such that*

$$\sup_{\mathbf{x} \in K} |f(\mathbf{x}) - P(\mathbf{x})| < \epsilon. \quad (9)$$

□

Theorem 2 indicates that continuous functions on a compact subset of $\mathbb{R}^n$ are approximated with polynomials. We next formulate the discrepancy of the two domains using the disagreement ratio and bound it with the label-wise moment divergence.

**Lemma 1.** *Let $\mathcal{D}$ and $\mathcal{D}'$ be two domains over an input space $\mathcal{X} \in \mathbb{R}^n$, where $n$ is the dimension of the inputs. Then, for any hypotheses $h_1, h_2 \in \mathcal{H}$ and any $\epsilon > 0$, there exist $n_\epsilon \in \mathbb{N}$ and a constant $a_{n_\epsilon}$ such that*

$$|\epsilon_{\mathcal{D}}(h_1, h_2) - \epsilon_{\mathcal{D}'}(h_1, h_2)| \leq \frac{1}{2} a_{n_\epsilon} \sum_{k=1}^{n_\epsilon} d_{LM,k}(\mathcal{D}, \mathcal{D}') + \epsilon. \quad (10)$$

□

*Proof.* Let the domains $\mathcal{D}$ and $\mathcal{D}'$ have the data distribution of $\mu(\mathbf{x})$ and $\mu'(\mathbf{x})$, respectively, over an input space $\mathcal{X}$, which is a compact subset of $\mathbb{R}^n$, where $n$ is the dimension of the inputs. For brevity, we denote $|\epsilon_{\mathcal{D}}(h_1, h_2) - \epsilon_{\mathcal{D}'}(h_1, h_2)|$ as $\Delta_{\mathcal{D}, \mathcal{D}'}$. Then,

$$\begin{aligned}
\Delta_{\mathcal{D}, \mathcal{D}'} &= |\epsilon_{\mathcal{D}}(h_1, h_2) - \epsilon_{\mathcal{D}'}(h_1, h_2)| \\
&\leq \sup_{h_1, h_2 \in \mathcal{H}} |\epsilon_{\mathcal{D}}(h_1, h_2) - \epsilon_{\mathcal{D}'}(h_1, h_2)| \\
&= \sup_{h_1, h_2 \in \mathcal{H}} \left| \Pr_{\mathbf{x} \sim \mu(\mathbf{x})} [h_1(\mathbf{x}) \neq h_2(\mathbf{x})] - \Pr_{\mathbf{x} \sim \mu'(\mathbf{x})} [h_1(\mathbf{x}) \neq h_2(\mathbf{x})] \right| \\
&= \sup_{h_1, h_2 \in \mathcal{H}} \left| \int_{\mathcal{X}} \mu(\mathbf{x}) \mathbf{1}_{h_1(\mathbf{x}) \neq h_2(\mathbf{x})} d\mathbf{x} - \int_{\mathcal{X}} \mu'(\mathbf{x}) \mathbf{1}_{h_1(\mathbf{x}) \neq h_2(\mathbf{x})} d\mathbf{x} \right|.
\end{aligned} \quad (11)$$

For any hypotheses $h_1, h_2$, the indicator function $\mathbf{1}_{h_1(\mathbf{x}) \neq h_2(\mathbf{x})}$ is Lebesgue integrable on $\mathcal{X}$, *i.e.* $\mathbf{1}_{h_1(\mathbf{x}) \neq h_2(\mathbf{x})}$ is a $L^1$ function. Since a set of continuous functions is dense in $L^1(\mathcal{X})$, for every

$\epsilon > 0$, there exists a continuous $L^1$ function $f$ defined on $\mathcal{X}$ such that

$$\left|\mathbf{1}_{h_1(\mathbf{x}) \neq h_2(\mathbf{x})} - f(\mathbf{x})\right| \leq \frac{\epsilon}{4} \tag{12}$$

for every $\mathbf{x} \in \mathcal{X}$, and the fixed $h_1$ and $h_2$ that drive equation 11 to the supremum. Accordingly,

$$f(\mathbf{x}) - \frac{\epsilon}{4} \leq \mathbf{1}_{h_1(\mathbf{x}) \neq h_2(\mathbf{x})} \leq f(\mathbf{x}) + \frac{\epsilon}{4}. \tag{13}$$

By integrating every term in the inequality over $\mathcal{X}$, the inequality,

$$\int_{\mathcal{X}} \mu(\mathbf{x}) f(\mathbf{x}) d\mathbf{x} - \frac{\epsilon}{4} \leq \int_{\mathcal{X}} \mu(\mathbf{x}) \mathbf{1}_{h_1(\mathbf{x}) \neq h_2(\mathbf{x})} d\mathbf{x} \leq \int_{\mathcal{X}} \mu(\mathbf{x}) f(\mathbf{x}) d\mathbf{x} + \frac{\epsilon}{4}, \tag{14}$$

follows. Likewise, the same inequality on the domain $\mathcal{D}'$ with $\mu'$ instead of $\mu$ holds. By subtracting the two inequalities and reformulating it, the inequality,

$$-\frac{\epsilon}{2} \leq \left|\left|\int_{\mathcal{X}} \mu(\mathbf{x}) \mathbf{1}_{h_1(\mathbf{x}) \neq h_2(\mathbf{x})} d\mathbf{x} - \int_{\mathcal{X}} \mu'(\mathbf{x}) \mathbf{1}_{h_1(\mathbf{x}) \neq h_2(\mathbf{x})} d\mathbf{x}\right| - \right.$$
$$\left.\left|\int_{\mathcal{X}} \mu(\mathbf{x}) f(\mathbf{x}) d\mathbf{x} - \int_{\mathcal{X}} \mu'(\mathbf{x}) f(\mathbf{x}) d\mathbf{x}\right|\right| \leq \frac{\epsilon}{2}, \tag{15}$$

is induced. By substituting the inequality in equation 15 to the equation 11,

$$\Delta_{\mathcal{D},\mathcal{D}'} \leq \left|\int_{\mathcal{X}} \mu(\mathbf{x}) f(\mathbf{x}) d\mathbf{x} - \int_{\mathcal{X}} \mu'(\mathbf{x}) f(\mathbf{x}) d\mathbf{x}\right| + \frac{\epsilon}{2}. \tag{16}$$

By the Theorem 2, there exists a polynomial $P(\mathbf{x})$ such that

$$\sup_{\mathbf{x} \in \mathcal{X}} |f(\mathbf{x}) - P(\mathbf{x})| < \frac{\epsilon}{4}, \tag{17}$$

and the polynomial $P(\mathbf{x})$ is expressed as

$$P(\mathbf{x}) = \sum_{k=1}^{n_\epsilon} \sum_{\mathbf{i} \in \Delta_k} \alpha_{\mathbf{i}} \prod_{j=1}^{n} (x_j)^{i_j}, \tag{18}$$

where $n_\epsilon$ is the order of the polynomial, $\Delta_k = \{\mathbf{i} = (i_1, \ldots, i_n) \in \mathbb{N}_0^n | \sum_{j=1}^n i_j = k\}$ is the set of the tuples of the nonnegative integers, which add up to $k$, $\alpha_{\mathbf{i}}$ is the coefficient of each term of the polynomial, and $\mathbf{x} = (x_1, x_2, \ldots, x_n)$. By applying equation 17 to the equation 16 and substituting the expression in equation 18,

$$\Delta_{\mathcal{D},\mathcal{D}'} \leq \left|\int_{\mathcal{X}} \mu(\mathbf{x}) P(\mathbf{x}) d\mathbf{x} - \int_{\mathcal{X}} \mu'(\mathbf{x}) P(\mathbf{x}) d\mathbf{x}\right| + \epsilon$$

$$= \left|\int_{\mathcal{X}} \mu(\mathbf{x}) \sum_{k=1}^{n_\epsilon} \sum_{\mathbf{i} \in \Delta_k} \alpha_{\mathbf{i}} \prod_{j=1}^{n} (x_j)^{i_j} d\mathbf{x} - \int_{\mathcal{X}} \mu'(\mathbf{x}) \sum_{k=1}^{n_\epsilon} \sum_{\mathbf{i} \in \Delta_k} \alpha_{\mathbf{i}} \prod_{j=1}^{n} (x_j)^{i_j} d\mathbf{x}\right| + \epsilon$$

$$\leq \sum_{k=1}^{n_\epsilon} \left|\sum_{\mathbf{i} \in \Delta_k} \alpha_{\mathbf{i}} \int_{\mathcal{X}} \mu(\mathbf{x}) \prod_{j=1}^{n} (x_j)^{i_j} d\mathbf{x} - \alpha_{\mathbf{i}} \int_{\mathcal{X}} \mu'(\mathbf{x}) \prod_{j=1}^{n} (x_j)^{i_j} d\mathbf{x}\right| + \epsilon$$

$$\leq \sum_{k=1}^{n_\epsilon} \sum_{\mathbf{i} \in \Delta_k} |\alpha_{\mathbf{i}}| \left|\int_{\mathcal{X}} \mu(\mathbf{x}) \prod_{j=1}^{n} (x_j)^{i_j} d\mathbf{x} - \int_{\mathcal{X}} \mu'(\mathbf{x}) \prod_{j=1}^{n} (x_j)^{i_j} d\mathbf{x}\right| + \epsilon$$

$$= \sum_{k=1}^{n_\epsilon} \sum_{\mathbf{i} \in \Delta_k} |\alpha_{\mathbf{i}}| \left|\int_{\mathcal{X}} \sum_{c \in \mathcal{C}} p(c) \mu_c(\mathbf{x}) \prod_{j=1}^{n} (x_j)^{i_j} d\mathbf{x} - \int_{\mathcal{X}} \sum_{c \in \mathcal{C}} p'(c) \mu'_c(\mathbf{x}) \prod_{j=1}^{n} (x_j)^{i_j} d\mathbf{x}\right| + \epsilon, \tag{19}$$

where $p(c)$ and $p'(c)$ are the probability that an arbitrary data is labeled as class $c$ in domain $\mathcal{D}$ and $\mathcal{D}'$, respectively, and $\mu_c(\mathbf{x}) = \mu(\mathbf{x}|y = c)$ and $\mu'_c(\mathbf{x}) = \mu'(\mathbf{x}|y = c)$ are the data distribution given

that the data is labeled as class $c$ on domain $\mathcal{D}$ and $\mathcal{D}'$, respectively. For $a_{\Delta_k} = \max_{\mathbf{i} \in \Delta_k} |\alpha_{\mathbf{i}}|$,

$$
\begin{aligned}
\Delta_{\mathcal{D},\mathcal{D}'} &\le \sum_{k=1}^{n_\epsilon} a_{\Delta_k} \sum_{\mathbf{i} \in \Delta_k} \left| \int_{\mathcal{X}} \sum_{c \in \mathcal{C}} p(c) \mu_c(\mathbf{x}) \prod_{j=1}^{n} (x_j)^{i_j} d\mathbf{x} - \int_{\mathcal{X}} \sum_{c \in \mathcal{C}} p'(c) \mu'_c(\mathbf{x}) \prod_{j=1}^{n} (x_j)^{i_j} d\mathbf{x} \right| + \epsilon \\
&\le \sum_{k=1}^{n_\epsilon} a_{\Delta_k} \sum_{\mathbf{i} \in \Delta_k} \sum_{c \in \mathcal{C}} \left| p(c) \int_{\mathcal{X}} \mu_c(\mathbf{x}) \prod_{j=1}^{n} (x_j)^{i_j} - p'(c) \int_{\mathcal{X}} \mu'_c(\mathbf{x}) \prod_{j=1}^{n} (x_j)^{i_j} \right| + \epsilon \\
&\le \sum_{k=1}^{n_\epsilon} a_{\Delta_k} d_{LM.k}(\mathcal{D}, \mathcal{D}') + \epsilon \\
&\le \frac{1}{2} a_{n_\epsilon} \sum_{k=1}^{n_\epsilon} d_{LM,k}(\mathcal{D}, \mathcal{D}') + \epsilon,
\end{aligned}
$$
(20)

for $a_{n_\epsilon} = 2 \max_{1 \le k \le n_\epsilon} a_{\Delta_k}$. □

Let $N$ datasets be drawn from $N$ labeled source domains $\mathcal{S}_1, \mathcal{S}_2, \ldots, \mathcal{S}_N$ respectively. We denote $i$-th source dataset $\{(\mathbf{x}_j^{\mathcal{S}_i}, y_j^{\mathcal{S}_i})\}_{j=1}^{n_{\mathcal{S}_i}}$ as $\mathbf{X}_{\mathcal{S}_i}$. The empirical error of hypothesis $h$ in $i$-th source domain $\mathcal{S}_i$ estimated with $\mathbf{X}_{\mathcal{S}_i}$ is formulated as $\hat{\epsilon}_{\mathcal{S}_i}(h) = \frac{1}{n_{\mathcal{S}_i}} \sum_{j=1}^{n_{\mathcal{S}_i}} \mathbf{1}_{h(\mathbf{x}_j^{\mathcal{S}_i}) \ne y_j^{\mathcal{S}_i}}$. Given a positive weight vector $\boldsymbol{\alpha} = (\alpha_1, \alpha_2, \ldots, \alpha_N)$ such that $\sum_{i=1}^{N} \alpha_i = 1$ and $\alpha_i \ge 0$, the weighted empirical source error is formulated as $\hat{\epsilon}_{\boldsymbol{\alpha}}(h) = \sum_{i=1}^{N} \alpha_i \hat{\epsilon}_{\mathcal{S}_i}(h)$.

**Lemma 2.** *For $N$ source domains $\mathcal{S}_1, \mathcal{S}_2, \ldots, \mathcal{S}_N$, let $n_{\mathcal{S}_i}$ be the number of samples from source domain $\mathcal{S}_i$, $m = \sum_{i=1}^{N} n_{\mathcal{S}_i}$ be the total number of samples from $N$ source domains, and $\boldsymbol{\beta} = (\beta_1, \beta_2, \ldots, \beta_N)$ with $\beta_i = \frac{n_{\mathcal{S}_i}}{m}$. Let $\epsilon_{\boldsymbol{\alpha}}(h)$ be the weighted true source error which is the weighted sum of $\epsilon_{\mathcal{S}_i}(h) = \Pr_{\mathbf{x} \sim \mu(\mathbf{x})}[h(\mathbf{x}) \ne y]$. Then,*

$$
\Pr\left[|\hat{\epsilon}_{\boldsymbol{\alpha}}(h) - \epsilon_{\boldsymbol{\alpha}}(h)| \ge \epsilon\right] \le 2 \exp\left(\frac{-2m\epsilon^2}{\sum_{i=1}^{N} \frac{\alpha_i^2}{\beta_i}}\right)
$$
(21)

*Proof.* It has been proven in Ben-David et al. (2010). □

We now turn our focus back to the Theorem 1 in the paper and complete the proof.

**Theorem 1.** *Let $\mathcal{H}$ be a hypothesis space of VC dimension $d$, $n_{\mathcal{S}_i}$ be the number of samples from source domain $\mathcal{S}_i$, $m = \sum_{i=1}^{N} n_{\mathcal{S}_i}$ be the total number of samples from $N$ source domains $\mathcal{S}_1, \ldots, \mathcal{S}_N$, and $\boldsymbol{\beta} = (\beta_1, \ldots, \beta_N)$ with $\beta_i = \frac{n_{\mathcal{S}_i}}{m}$. Let us define a hypothesis $\hat{h} = \arg\min_{h \in \mathcal{H}} \hat{\epsilon}_{\boldsymbol{\alpha}}(h)$ that minimizes the weighted empirical source error, and a hypothesis $h_{\mathcal{T}}^* = \arg\min_{h \in \mathcal{H}} \epsilon_{\mathcal{T}}(h)$ that minimizes the true target error. Then, for any $\delta \in (0, 1)$ and $\epsilon > 0$, there exist $N$ integers $n_\epsilon^1, \ldots, n_\epsilon^N$ and $N$ constants $a_{n_\epsilon^1}, \ldots, a_{n_\epsilon^N}$ such that*

$$
\epsilon_{\mathcal{T}}(\hat{h}) \le \epsilon_{\mathcal{T}}(h_{\mathcal{T}}^*) + \eta_{\boldsymbol{\alpha},\boldsymbol{\beta},m,\delta} + \epsilon + \sum_{i=1}^{N} \alpha_i \left(2\lambda_i + a_{n_\epsilon^i} \sum_{k=1}^{n_\epsilon^i} d_{LM,k}(\mathcal{S}_i, \mathcal{T})\right)
$$
(22)

*with probability at least $1 - \delta$, where $\eta_{\boldsymbol{\alpha},\boldsymbol{\beta},m,\delta} = 4\sqrt{\left(\sum_{i=1}^{N} \frac{\alpha_i^2}{\beta_i}\right)\left(\frac{2d\left(\log\left(\frac{2m}{d}\right)+1\right)+2\log\left(\frac{4}{\delta}\right)}{m}\right)}$ and $\lambda_i = \min_{h \in \mathcal{H}} \{\epsilon_{\mathcal{T}}(h) + \epsilon_{\mathcal{S}_i}(h)\}$.* □

*Proof.*

$$
|\epsilon_{\boldsymbol{\alpha}}(h) - \epsilon_{\mathcal{T}}(h)| = \left|\sum_{i=1}^{N} \alpha_i \epsilon_{\mathcal{S}_i}(h) - \epsilon_{\mathcal{T}}(h)\right| \le \sum_{i=1}^{N} \alpha_i |\epsilon_{\mathcal{S}_i}(h) - \epsilon_{\mathcal{T}}(h)|.
$$
(23)

We define $h_i^* = \arg\min_{h \in \mathcal{H}} \epsilon_{\mathcal{S}_i}(h) + \epsilon_{\mathcal{T}}(h)$ for every $i = 1, 2, \ldots, N$ for the following equations. We also note that the 1-triangular inequality (Crammer et al. (2008)) holds for binary classification

tasks, *i.e.*, $\epsilon_{\mathcal{D}}(h_1, h_2) \leq \epsilon_{\mathcal{D}}(h_1, h_3) + \epsilon_{\mathcal{D}}(h_2, h_3)$ for any hypothesis $h_1, h_2, h_3 \in \mathcal{H}$ and domain $\mathcal{D}$. Then,

$$|\epsilon_{\mathcal{D}}(h) - \epsilon_{\mathcal{D}}(h, h')| = |\epsilon_{\mathcal{D}}(h, l_{\mathcal{D}}) - \epsilon_{\mathcal{D}}(h, h')| \leq \epsilon_{\mathcal{D}}(l_{\mathcal{D}}, h') = \epsilon_{\mathcal{D}}(h') \tag{24}$$

for the ground truth labeling function $l_{\mathcal{D}}$ on the domain $\mathcal{D}$ and two hypotheses $h, h' \in \mathcal{H}$. Applying the definition and the inequality to equation 23,

$$
\begin{aligned}
|\epsilon_{\boldsymbol{\alpha}}(h) - \epsilon_{\mathcal{T}}(h)| &\leq \sum_{i=1}^{N} \alpha_i \left( |\epsilon_{\mathcal{S}_i}(h) - \epsilon_{\mathcal{S}_i}(h, h_i^*)| + |\epsilon_{\mathcal{S}_i}(h, h_i^*) - \epsilon_{\mathcal{T}}(h, h_i^*)| + |\epsilon_{\mathcal{T}}(h, h_i^*) - \epsilon_{\mathcal{T}}(h)| \right) \\
&\leq \sum_{i=1}^{N} \alpha_i \left( \epsilon_{\mathcal{S}_i}(h_i^*) + |\epsilon_{\mathcal{S}_i}(h, h_i^*) - \epsilon_{\mathcal{T}}(h, h_i^*)| + \epsilon_{\mathcal{T}}(h_i^*) \right)
\end{aligned}
\tag{25}
$$

By the definition of $h_i^*$, $\epsilon_{\mathcal{S}_i}(h_i^*) + \epsilon_{\mathcal{T}}(h_i^*) = \lambda_i$ for $\lambda_i = \min_{h \in \mathcal{H}} \{\epsilon_{\mathcal{T}}(h) + \epsilon_{\mathcal{S}_i}(h)\}$. Additionally, according to Lemma 1, for any $\epsilon > 0$, there exists an integer $n_\epsilon$ and a constant $a_{n_\epsilon^i}$ such that

$$|\epsilon_{\mathcal{S}_i}(h, h_i^*) - \epsilon_{\mathcal{T}}(h, h_i^*)| \leq \frac{1}{2} a_{n_\epsilon^i} \sum_{k=1}^{n_\epsilon^i} d_{LM,k}(\mathcal{S}_i, \mathcal{T}) + \frac{\epsilon}{2}. \tag{26}$$

By applying these relations,

$$
\begin{aligned}
|\epsilon_{\boldsymbol{\alpha}}(h) - \epsilon_{\mathcal{T}}(h)| &\leq \sum_{i=1}^{N} \alpha_i \left( \lambda_i + \frac{1}{2} a_{n_\epsilon^i} \sum_{k=1}^{n_\epsilon^i} d_{LM,k}(\mathcal{S}_i, \mathcal{T}) + \frac{\epsilon}{2} \right) \\
&\leq \sum_{i=1}^{N} \alpha_i \left( \lambda_i + \frac{1}{2} a_{n_\epsilon^i} \sum_{k=1}^{n_\epsilon^i} d_{LM,k}(\mathcal{S}_i, \mathcal{T}) \right) + \frac{\epsilon}{2}.
\end{aligned}
\tag{27}
$$

By Lemma 2 and the standard uniform convergence bound for hypothesis classes of finite VC dimension (Ben-David et al. (2010)),

$$
\begin{aligned}
\epsilon_{\mathcal{T}}(\hat{h}) &\leq \epsilon_{\boldsymbol{\alpha}}(\hat{h}) + \frac{\epsilon}{2} + \sum_{i=1}^{N} \alpha_i \left( \lambda_i + \frac{1}{2} a_{n_\epsilon^i} \sum_{k=1}^{n_\epsilon^i} d_{LM,k}(\mathcal{S}_i, \mathcal{T}) \right) \\
&\leq \hat{\epsilon}_{\boldsymbol{\alpha}}(\hat{h}) + \frac{1}{2} \eta_{\boldsymbol{\alpha},\boldsymbol{\beta},m,\delta} + \frac{\epsilon}{2} + \sum_{i=1}^{N} \alpha_i \left( \lambda_i + \frac{1}{2} a_{n_\epsilon^i} \sum_{k=1}^{n_\epsilon^i} d_{LM,k}(\mathcal{S}_i, \mathcal{T}) \right) \\
&\leq \hat{\epsilon}_{\boldsymbol{\alpha}}(h_{\mathcal{T}}^*) + \frac{1}{2} \eta_{\boldsymbol{\alpha},\boldsymbol{\beta},m,\delta} + \frac{\epsilon}{2} + \sum_{i=1}^{N} \alpha_i \left( \lambda_i + \frac{1}{2} a_{n_\epsilon^i} \sum_{k=1}^{n_\epsilon^i} d_{LM,k}(\mathcal{S}_i, \mathcal{T}) \right) \\
&\leq \epsilon_{\boldsymbol{\alpha}}(h_{\mathcal{T}}^*) + \eta_{\boldsymbol{\alpha},\boldsymbol{\beta},m,\delta} + \frac{\epsilon}{2} + \sum_{i=1}^{N} \alpha_i \left( \lambda_i + \frac{1}{2} a_{n_\epsilon^i} \sum_{k=1}^{n_\epsilon^i} d_{LM,k}(\mathcal{S}_i, \mathcal{T}) \right) \\
&\leq \epsilon_{\mathcal{T}}(h_{\mathcal{T}}^*) + \eta_{\boldsymbol{\alpha},\boldsymbol{\beta},m,\delta} + \epsilon + \sum_{i=1}^{N} \alpha_i \left( 2\lambda_i + a_{n_\epsilon^i} \sum_{k=1}^{n_\epsilon^i} d_{LM,k}(\mathcal{S}_i, \mathcal{T}) \right).
\end{aligned}
\tag{28}
$$

The last inequality holds by equation 27 with $h = h_{\mathcal{T}}^*$. $\qquad\square$

