# OpenReview forum: "Multi-EPL: Accurate Multi-source Domain Adaptation"
_ICLR.cc/2021/Conference — Reject_

### Official Review · AnonReviewer1 · 2020-10-26
**Official Blind Review #1**

**Rating:** 4
**Confidence:** 5

**Review:**

In this paper, the authors propose a MULTI-EPL for multi-source domain adaptation. The key idea includes two folds: (1) to align label-wise moment, and (2) to ensemble multiple feature extractor. Experimental studies on 3 datasets are done to verify the proposed MUTL-EPL.

Overall, the paper is well-written. The technical approach is simple and sound. My major concern is on the technical significance of the method. Here are the detailed comments:

(1)	One motivation of the paper is that current methods fail to consider the shifts among sources. However, there are some multiple source transfer methods explicitly modelling the inter-domain similarities, e.g., [ref1].  The paper also misses some important multiple-source references. Please refer to the survey [ref2] for different types of multiple source transfer methods. It would be better to have a comprehensive discussion on the related works.

(2)	The proposed label-wise moment matching is not new in transfer. Early subspace based work, e.g., JDA [ref3], and latest semantic deep learning based transfer methods, e.g., [ref4] share the similar idea.

(3)	The threshold \taw is used to obtain the good target labels.  On the one hand, it is unclear how this parameter should be set for different transfer tasks. On the other hand, high confidence score does not imply correct target label prediction. Error reinforcement may happen even with a well-tuned \taw.

(4)	The usage of ensemble feature extractor is actually using high complexity to enhance prediction accuracy. The scalability could be an issue of the proposed method, especially considering that multiple source may have extremely large data size.

(5)	Sensitivity analyses on the balancing parameters \alpha and \beta should be done.

(6)	Why data augmentation is done on office-caltech10 datasets? There are many datasets containing multiple domains with sufficient data, e.g., office-home, Please use these datasets instead of constructing `artificial’ real-world dataset.

(7)	The baseline methods can be further improved (please consider [ref2] for more baselines). Regarding the current results, MUTLT-EPL performs larger improvements over M3SDA on the 2nd, 3rd, and 4th tasks in Digit –five dataset, while only achieves marginal improvements in other tasks, e.g.. tasks in office-caltech10 dataset. More analyses on the performance difference of different tasks should be discussed.

(8)	Based on the ablation study, Mutil-epl-r achieves comparable results with multi-epl, which indicates that the extractor classifier and feature diversifying loss have less importance in the overall objective.

[ref1] Source-target similarity modelings for multi-source transfer gaussian process regression

[ref2] Multi-source Domain Adaptation in the Deep Learning Era: A Systematic Survey

[ref3] Transfer Feature Learning with Joint Distribution Adaptation

[ref4] Deep Transfer Learning with Joint Adaptation Networks

Update: Thanks for the authors' response. However, I am not convinced on several points, e.g., (3) - (7). Considering the other reviewers' comments, I think the paper needs to be further improved. Thus, I will keep my score.

---

> ### Author Response · Authors · 2020-11-24
> **We appreciate your concern toward our work.**
>
> We appreciate your concern toward our work. Thanks to your review, we revised our paper to answer your concerns.
> 1.	The paper misses some important multiple-source references.
> - We added additional reference to the related work section.
>
> 2.	The proposed label-wise moment matching is not new in transfer.
> - The existing methods that have similar idea with ours have different focuses. Therefore, they should be distinguished from our approach.
>
> 3.	The threshold tau is used to obtain the good target labels. On the one hand, it is unclear how this parameter should be set for different transfer tasks. On the other hand, high confidence score does not imply correct target label prediction. Error reinforcement may happen even with a well-tuned tau.
> - High confidence does not guarantee correct target label prediction but we think that there is a strong correlation between confidence and correctness.
>
> 4.	The usage of ensemble feature extractor is actually using high complexity to enhance prediction accuracy. The scalability could be an issue of the proposed method, especially considering that multiple source may have large data size.
> - Even without ensemble, our method still shows the superiority. Thus, we believe that the model without ensemble learning can be used if there are resource constraint issues.
>
> 5.	Sensitivity analysis on the balancing parameters alpha and beta should be done.
> - Because of the time limitation, we could not conduct the sensitivity analysis thoroughly. However, we observed that reducing alpha to 0.0001 in Multi-EPL (n=2) brings great performance degradation: from 83.95 to 78.16 with MNIST-M as the target domain.
>
> 6.	Why data augmentation is done on office-caltech10 datasets? There are many datasets containing multiple domains with sufficient data. Please use these datasets instead of constructing artificial real-world dataset.
> - OFFICE-Caltech10 is one of the mostly used datasets for MSDA thanks to its domain diversity and imbalance (the number of data differs greatly depending on the domain). Data augmentation cannot be stated as the pitfall of OFFICE-Caltech dataset since the number of the data per label is the same as OFFICE31 or OFFICE-Home dataset.
>
> 7.	More analyses on the performance difference of different tasks should be discussed.
> - For OFFICE-Caltech10 dataset, the state-of-the-art performance is already very high. It is the reason for the marginal improvements. Our method made relatively great enhancement compared to the other methods’ enhancement.
>
> 8.	Based on the ablation study, Multi-EPL-R achieves comparable results with Multi-EPL, which indicates that the extractor classifier and feature diversifying loss have less importance in the overall objective.
> - As you have stated, extractor classifier and feature diversity loss are proved to be unnecessary for the model. It shows that we can utilize ensemble learning method without any concern about the redundancy in the features.

---

### Official Review · AnonReviewer3 · 2020-10-27
**Although the design of the proposed method seems reasonable, its novelty is marginal.**

**Rating:** 4
**Confidence:** 4

**Review:**

--Paper summary--

The authors propose a novel method for multi-source domain adaptation (MSDA). For effective adaptation, the proposed method adopts three techniques: (1) label-wise moment matching, (2) pseudo-labeling target data, and (3) ensembling multiple feature extractors. Experimental results show that the proposed method outperforms several state-of-the-art methods in both image and text domains.

--Review summary--

Although the design of the proposed method seems reasonable, its novelty is marginal. Additionally, the evaluation in the experiments is somewhat unfair. I vote for rejection.

--Details--

Strength

- This paper is well-organized and is easy to follow. I believe that the proposed method can be easily implemented without any obstacles.
- Good performance in both image and text domains is appealing. Such results should be highly appreciated especially in machine learning community.

Weakness and concerns

- Marginal novelty. The three techniques that the proposed method adopted are all similar to those already proposed in the literature. I could not find any novel and specific design or strategy to combine them specialized for MSDA.
	- Class-wise distributional alignment is a common technique in recent domain adaptation methods, e.g. [R1] and [R2].
[R1] "A DIRT-T Approach to Unsupervised Domain Adaptation," ICLR 2018.
[R2] "Unsupervised Domain Adaptation via Regularized Conditional Alignment," ICCV 2019.
	- Pseudo-labeling is also a common technique in recent domain adaptation methods, e.g. [R1] and [R3].
[R3] "Asymmetric Tri-training for Unsupervised Domain Adaptation," ICML 2017.
	- Using multiple feature representations is not so common but is presented in [R4] and [R5].
[R4] "Domain Adaptation with Ensemble of Feature Groups," IJCAI 2011.
[R5] "Domain Separation Networks," NeurIPS 2016.
- The design of the feature diversifying loss is not reasonable. It can be minimized by just making feature representations to be easy to discrminate their extractors, which does not necessarily increase the diversity of the representations. For example, given two extractors that share the same parameters, adding a large offset to outputs from one extractor leads to high performance of the extractor classifier but does not increase diversity of the feature representations.
- The exprimental setting is somewhat unfair. Since the proposed method utilizes n feature extractors, the model complexity in the proposed method should be n times larger than that in existing methods.

---

> ### Author Response · Authors · 2020-11-24
> **Thank you for the advices on our paper.**
>
> Thank you for the advices on our paper. We have considered your concern for a couple of weeks and summarized our thoughts below.
>
> 1.	Marginal novelty. The three techniques that the proposed method adopted are all similar to those already proposed in the literature. I could not find any novel and specific design or strategy to combine them specialized for MSDA.
> - Even though some of the approaches in our paper resemble the other papers’ idea, applying those strategies for MSDA is newly proposed in this paper.
>
> 2.	The design of the feature diversifying loss is not reasonable. It can be minimized by just making feature representations to be easy to discriminate their extractors, which does not necessarily increase the diversity of the representation. For example, given two extractors that share the same parameters, adding a large offset to outputs from one extractor leads to high performance of the extractor classifier but does not increase diversity of the feature representations.
> - We agree that the design of the feature diversifying loss is unreasonable. Nonetheless, we wished to point out that using ensemble method without any efforts to diversify the features still provides great performance enhancement.
>
> 3.	The experimental setting is somewhat unfair. Since the proposed method utilizes n feature extractors, the model complexity in the proposed method should be n times larger than that in existing methods.
> - By comparing our method with only 1 feature extractor, we have verified the effectiveness of our method in a fair setting. Also, there are many cases that increasing model complexity does not improve the performance. Thus, our method to increase the model complexity in effective way seems to be convincing.

---

### Official Review · AnonReviewer2 · 2020-10-28
**Limited novelty**

**Rating:** 4
**Confidence:** 5

**Review:**

- Summary and contributions
    - In this work, the authors proposed an algorithm for multi-source domain adaptation. While the results seem promising, the technical contribution is incremental and limited. Meanwhile, more empirical results are needed to validate the effectiveness of the framework.
- Strengths:
    - The paper is well written and easy to follow.
    - The problem investigated in this paper, i.e., multi-source domain adaptation, is of significance.
- Weaknesses:
    - The technical contribution of this work is limited. Label-wise moment matching or adversarial training (e.g., [1]) has been a common practice in single-source domain adaptation. The authors simply applies this idea of multi-mode aware domain adaptation to multi-source domain adaptation. Moreover, comparing the first line, i.e., MULTI-0, in Table 2 with M3SDA in Table 1(a), we find that this label-wise moment-matching makes almost no contribution.
    - The empirical results, especially the ablation studies, do not hold. Pseudo-labeling the unlabeled data in the target domain and using multiple feature extractors can also be easily used by M3SDA and DCTN. Expecting a definite performance boost, I expect the authors to provide such results. Moreover, pseudo-labeling and ensemble learning, however, are not novel; they are widely adopted techniques and can be easily incorporated into any algorithm, e.g., M3SDA, for performance improvement.
    - Even on the dataset Office-Caltech and Amazon Reviews, the performance improvement of the proposed algorithm is minor.


[1] Pei, Z., Cao, Z., Long, M., & Wang, J. (2018). Multi-adversarial domain adaptation. AAAI 2018.

---

> ### Author Response · Authors · 2020-11-24
> **We appreciate your comprehensive review.**
>
> We appreciate your comprehensive review. Here are the responses for your concerns.
> 1.	The technical contribution of this work is limited. Label-wise moment matching or adversarial training has been a common practice in single-source domain adaptation.
> - Even though those methods have been exploited for single-source domain adaptation, introducing the strategies to multi-source domain adaptation and enhancing the performance is meaningful.
>
> 2.	Comparing the first line, i.e., Multi-0, in Table 2 with M3SDA in Table 1(a), we find that this label-wise moment matching makes almost no contribution.
> - Multi-0, which aligns moments regardless of the labels, is just the same as M3SDA so it is natural that M3SDA and Multi-0 have the same performance. In order to examine the effectiveness of label-wise moment matching, you should compare Multi-0 and Multi-PL.
>
> 3.	The empirical results, especially the ablation studies, do not hold. Pseudo-labeling the unlabeled data in the target domain and using multiple feature extractors can also be easily used by M3SDA and DCTN. Expecting a definite performance boost, I expect the authors to provide such results.
> - As you have mentioned, our methods can be easily used along with the other strategies. We believe it is the strength of our method not the weakness.
>
> 4.	Even on the dataset Office-Caltech and Amazon Reviews, the performance improvement of the proposed algorithm is minor.
> - Compared to the performance enhancement that the existing methods have made, the performance improvement that our algorithm has made cannot be underestimated.

---

### Official Review · AnonReviewer4 · 2020-10-29
**Easy to read, but unclear on a few points.**

**Rating:** 5
**Confidence:** 3

**Review:**

This paper studies the multi-source domain adaptation (MSDA) problem. The authors argue that the existing MSDA solutions (1) do not explicitly consider distribution conditioned on labels of each domain, (2) rely on limited feature extraction based on one extractor, (3) do not well explore target data due to the absence of label. Correspondingly, Multi-EPL is proposed based on moment matching.

Although the design of the proposed method seems reasonable, its novelty is marginal. Additionally, the evaluation in the experiments is somewhat unfair. I vote for rejection.

The paper is well organized. The technical details are clearly presented. A few comments are summarized below.
-In Eq.1, the authors minimize the discrepancy between every two distinct domains. It is unclear to me why not minimize the pairs of each source domain and target domain, (N-pairs). The goal is to align the distributions between source and target. Please clarify the motivation of aligning two source domains.

Besides, it would be good to have one baseline only considering label-wise moment matching losses for only between N source and 1 target pairs.

-In page 5, the motivation of diversifying features from different extractors is unclear to me. Please clarify the benefit of classifying feature according to extractor ID labels. Moreover, the ablation study presented in section 5.3, does not show a clear improvement by introducing the diversifying loss. I would encourage the authors to design another analytical experiment to show its effectiveness.

-The performance improvement compared to the state of the arts is limited. Specifically, for Digit-Five dataset, a missing recent work [ref-1] reports an average performance of 91.8. To show the consistent performance improvement over this strong baseline, I’d encourage the authors cite and compare it under the same setting.
[ref-1] Hang Wang et al., Learning to Combine: Knowledge Aggregation for Multi-Source Domain Adaptation, ECCV 2020

-Besides [ref-1], there are several other recent MSDA papers are missing, including but not limited to
[ref-2] Chuang Lin et al., Multi-source Domain Adaptation for Visual Sentiment Classification, https://arxiv.org/abs/2001.03886
[ref-3] Haotian Wang et al., Tmda: Task-specific multi-source domain adaptation via clustering embedded adversarial training. ICDM 2019.

-Other minor point. A typo in page 7, “threshoold” -> “threshold”

Updates: Thanks for the authors' response. Some of my queries (1st and 3rd) were clarified. However, unfortunately, I still think more needs to be done to show the superiority of the results. I retain my original decision.

---

> ### Author Response · Authors · 2020-11-24
> **Thank you for the reviews on my paper.**
>
> 1. In Eq.1, the authors minimize the discrepancy between every two distinct domains. It is unclear to me why not minimize the pairs of each source domain and target domain (N-pairs). The goal is to align the distributions between the source and target. Please clarify the motivation of aligning two source domains. Besides, it would be good to have one baseline only considering label-wise moment matching losses for only between N source and 1 target pairs.
> - Aligning two source domains allows the feature extractor to be more generalized by providing more guidance to be trained. Moreover, as we align features in a label-wise manner while exploiting pseudo-labels for the target data that are not guaranteed to be correct, aligning two source domains resolves the bad adaptation attributed to the wrongly labeled target data.
>
> 2.	In page 5, the motivation of diversifying features form different extractors is unclear to me. Please clarify the benefit of classifying feature according to extractor ID labels. Moreover, the ablation study presented in section 5.3, does not show a clear improvement by introducing the diversifying loss. I would encourage the authors to design another analytical experiment to show its effectiveness.
> - If the two features from the two extractors are the same, then ensemble will be meaningless. Thus, we thought that diversifying features from the two different feature extractors would increase the knowledge in the ensembled features. As you have stated, we fully agree that newly designed experiment is essential to prove its effectiveness regardless its motivation.
>
> 3.	The performance improvement compared to the state of the arts is limited. Specifically, for Digit-Five dataset, a missing recent work reports an average performance of 91.8. To show the consistent performance improvement over this strong baseline, I’d encourage the authors cite and compare it under the same setting. Besides, there are several other recent MSDA papers are missing.
> - We could not reflect all the recent works while conducting the experiments and we appreciate you for letting us know about the novel methods. In order to fit the deadline of rebuttal, we could not additionally conduct the experiments, but we added them to the related works so that readers can compare the two methods and get to know the superiority of our method.
>
> 4.	Typo in page 7: We fixed the typo.

---

### Decision · Program_Chairs · 2021-01-07
**Final Decision**

**Decision:**

Reject

**Comment:**

In this paper, the authors proposed a solution to the problem of multi-source domain adaptation. All the reviewers have two concerns: 1) the technical contribution/novelty is limited, and 2) the experimental results are not convincing. Therefore, this paper does not meet the standard of being published in ICLR. The authors are encouraged to improve this work by addressing the issues raised by the reviewers.